# City Water Resource Allocation Considering Energy Consumption in Jinan, China

**Zhaohui Yang [1,2,\*], G. Mathias Kondolf [2], Jie Du [3] and Luyao Cai [1]**

1   Department of Water Resources, China Institute of Water Resources and Hydropower Research, Beijing 100038, China; cailuyao_cc@163.com
2   Department of Landscape Architecture & Environmental Planning, University of California Berkeley, Berkeley, CA 94704, USA; kondolf@berkeley.edu
3   Power China Chengdu Engineering Co., Ltd., Chengdu 610072, China; taiyuandujie1992@163.com
\*   Correspondence: yzh2010@iwhr.com; Tel.: +86-137-0138-1419

**Abstract:** The conflict between urban energy supply and demand is becoming increasingly evident. One aspect that consumes a great deal of this energy is the allocation of urban water resources. This study proposes a new scheme for rationally allocating urban water resources considering the high levels of energy currently consumed in Jinan city of Shandong, China. The focus is on simultaneously minimizing energy consumption and water shortage rates and granting priority to public water supplies in line with the predicted water supply levels for all available sources. Based on this assessment, further adjustments were made in terms of system configuration and the analysis of energy consumption. The results of the general water resource allocation model not only show that Jinan's total water supply in 2030 will increase by 33.7% from 2019 but that energy consumption will also increase by 58.5%. If energy consumption is constrained and water supplies are restricted for high-energy-consumption activities, the results of the water resource allocation model considering energy consumption show that energy consumption will increase only by 44.2%. And the results also show that local groundwater is less energy intensive than imported surface water, which suggests that groundwater should be preferred (at least for energy reasons). Through modeling to reduce the total energy consumption in water resource allocation, this paper can provide a reference for energy saving for urban water supply systems.

**Keywords:** water resource allocation; water extraction and treatment; energy consumption analysis; water supply

## 1. Introduction

American scientists have great paid attention to energy consumption in water resource utilization [1–6]. The water supply sector is one of the largest energy loads in most municipal regions [7–9]. Water-related services such as water supplies, sewage treatment, and others account for 19% of the electricity used in California. Initiated by the American Electric Power Association in 2000, a series of studies on water resources and sustainable development were carried out, including statistics on the supply of public water resources, electricity consumption for sewage treatment by users, and the prediction of water consumption and electricity consumption by users [10]. Nationally, they showed that direct water-related energy consumption (i.e., energy considered in the direct water services and the direct steam use categories) was approximately 12.6% (12.3 ± 0.346 quads) of the 2010 national primary energy consumption [11].

The price of electricity required to produce and transport water influences the cost of water supply options and may alter the decision makers' economic ranking of these options [12]. Quantifying energy consumption in the process of water resource utilization has become a hot research topic, such as studying the relationship between capital, economy, water, and energy [13]. Researchers have summarized the water consumption

of each link of energy utilization and the water consumption of different power generation methods [14,15]. Researchers have also quantified energy consumption in the water resource utilization cycle and water consumption in the process of energy utilization [16] and the impact of different water source options on water supply costs, energy consumption, and greenhouse gas emissions. A city's current water supply was examined, and researchers estimated its future energy requirements based on water supply projections [17] and quantified urban water-related energy use [18].

From 1978 to 2015, China's population of permanent urban residents increased from 170 million to 770 million, and the urbanization rate steadily climbed from 17.9% to 56.1%. For each 1.35% rise in the urbanization rate, total and per capita energy consumption expenses increased by 9.5% and 8.5%. The pressure on the available urban energy supply is increasing as well as China's domestic energy demand. Moreover, China has now become the world's largest consumer of energy and accounts for 23 percent of global energy consumption; thus, the issue of protecting and conserving urban energy supplies has become a major unavoidable issue regarding China's urbanization.

Reducing the amount of energy consumed in allocating urban water resources will greatly contribute toward these goals and have important practical significance. At present, it is a difficult research point to analyze and quantify the power consumption of each link of the social water cycle, such as supply, use, consumption, drainage, and treatment, and the constraint threshold of energy consumption on the water supply of different structures. As the conflict between urban energy supplies and demand is becoming increasingly evident, we should be managing water resources to account for saving energy consumption, and exploring methods of urban water resource allocation based on energy consumption represents a new approach in this field.

## 2. Study Area and Data

### 2.1. Study Area

Jinan city is the capital of China's Shandong Province. It has many natural springs and is known as the "Spring City". The location of Jinan city is shown in Figure 1. This city has a unique location and many rivers. Some areas are in the Yellow River Basin, some are in the Haihe River Basin, and some are in the Huaihe River Basin. The city has a limited water supply, with existing water sources mainly including surface water, groundwater, reclaimed water, and external water transfers (including water sources from the Yellow River and Yangtze River), see Table 1. It is the use of inter-basin water transfers from locations outside of Jinan that has allowed for its high population rate and economic growth. The inherent conflict between maintaining a steady water supply and maintaining the health of the local springs is highly evident. The issue of electricity consumption is also extremely severe. There have been many large-scale urban power cuts in Jinan, and its overall energy situation is strongly representative of cities throughout northern China.

**Table 1.** Stressors on the sustainability of Jinan city's water supply sources.

| Sources | Stressors | | | | |
| --- | --- | --- | --- | --- | --- |
| | Competing Demands | Climate Change | Water Quality Constraints | Energy Constraints | Cost Constraints |
| Surface water | ▪ | ▪ | | ▪ | ▪ |
| Groundwater | | | ▪ | ▪ | ▪ |
| Reclaimed water | | | ▪ | ▪ | ▪ |
| Yellow River water | ▪ | ▪ | ▪ | ▪ | ▪ |
| Yangtze River water | | ▪ | | ▪ | ▪ |

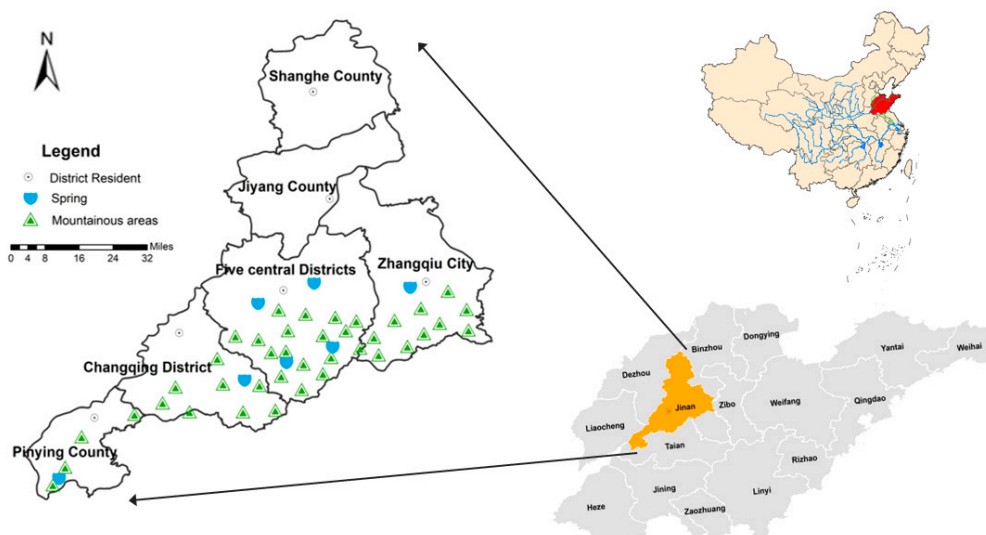

**Figure 1.** Location of Jinan city.

*2.2. Data Description*

Total water consumption, industrial water consumption, agricultural water consumption, domestic water consumption, ecological water consumption, and other water resources data were mainly from the Jinan Water Resources Statistical Bulletin [19]. The data on water conservancy facilities are from the China Water Conservancy Yearbook [20]. The population, urbanization rate, GDP, and industrial production value data are from the Jinan Statistical Yearbook [21]. The energy intensity data come from the BP Statistical Yearbook of World Energy [22] and the China Energy Statistics Yearbook [23]. Jinan Water Supply Group and Jinan Guangda Sewage Co., Ltd. also provide the energy consumption status of water supply, use, consumption, discharge, and treatment in Jinan.

To implement strict WRM, each province, prefecture, and county has the constraint of "Three Red Lines" on water resources management. This paper uses the median value of the estimated water demand from the Comprehensive Water Resources Plan for Jinan City and the Comprehensive Water Resources Plan for Shandong Province as the threshold for future total water use and consumption goals. A geographic breakdown of this goal and the total threshold is shown in Figure 2.

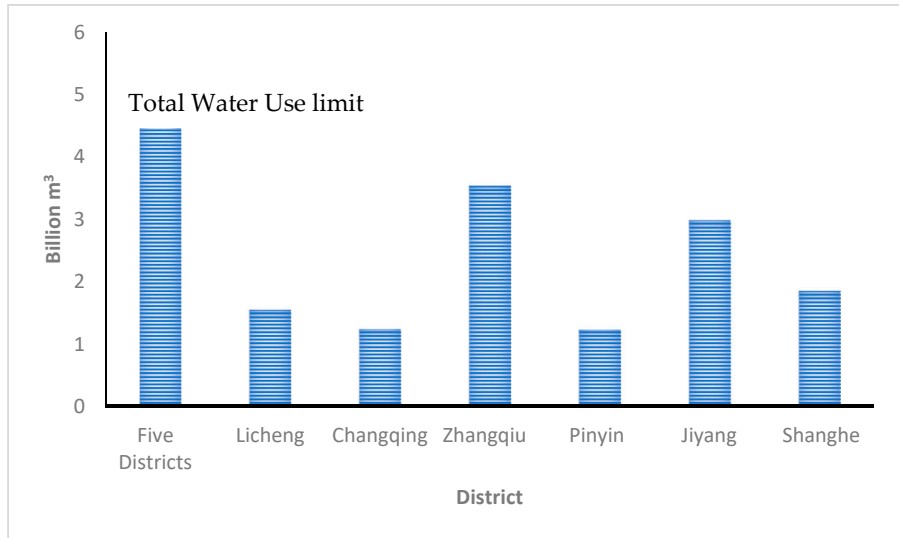

**Figure 2.** Recommended control indicators for total water utilization in all districts and counties of Jinan city in 2030.

## 3. Research Methodology

There are several aspects to the issue of energy consumption in allocating urban water resources. The first phase is the extraction and transportation of raw water. Most water sources (including surface water, groundwater, seawater, reclaimed water, and externally transferred water) require pumping stations to transport raw water to the processing stations. There, they enter the second phase: water production. In conventional waterworks, water is purified by methods such as sedimentation, filtration, and disinfection. According to our survey, tracking the energy performance of the Jinan water system, most of the energy consumption in this phase occurs during the pumping and treatment processes; the water pump consumes about 90% of the electricity used in this phase, while cleaning and sewage discharge account for about 10%. The third phase is the distribution of the treated water. Tap water distribution systems deliver treated water to users through booster pumps. Most of the energy consumed in this phase is used in pumping. The fourth phase is the collection, treatment, and discharge of sewage. The main sources of energy consumption in this phase include pumping stations, fans, biochemical treatment, and other related treatment methods. Both the quality and quantity of the water will have an impact on the unit energy consumption of each water processing mechanism, and the energy consumption patterns of different water supply infrastructures are all different. Due to limited data availability, this paper considers only the energy consumption of the water intake and production processes; that is, it analyzes only the pumping, transportation, treatment, and processing of water, as well as the selection of urban water sources and their energy consumption levels.

Through the quantitative analysis of energy consumption, the energy consumption evaluation function can be established, the advantages and disadvantages of various water sources can be identified, and the water supply thresholds of various water sources can be analyzed. Based on the hydrological process control equation, a water cycle simulation model is constructed; referring to the existing research results, a socio-economic water demand prediction simulation module and available water quantity prediction module are established; at the same time, on the basis of the existing configuration model, the water per 10,000 cubic meters is the basic module. According to the target of the lowest overall energy consumption in each link of supply, use, consumption, discharge, and treatment, an energy consumption module is established, and the overall lowest artificial energy consumption is taken as the constraint of the model, and the water supply prediction module of the configuration model is entered. The water resource allocation model is used to analyze the balance of water supply and demand.

### 3.1. Principles of Configuration

The principle of minimum human consumption of energy is paramount considering all aspects of urban water resource utilization. When allocating water resources, this goal is to be maintained, while also considering the risk characteristics of each water source, its potential water supply, and the intensity of energy consumption. The consumption threshold is taken as a configuration constraint, and local water resources are prioritized for utilization. However, in areas where groundwater has been over-exploited, the scale and layout of such exploitation should be appropriately reduced and adjusted in the future.

### 3.2. Objective Function

The objective function of the configuration model is used to evaluate the advantages and disadvantages of different experimental operational modes and thus review the results of all possible operational variations and find the solution that minimizes or maximizes the objective function. The objective function in this model considers mainly how to achieve the greatest positive effect on the water supply with the minimum loss of water. It is limited by constraints such as the minimum flow of local rivers required to maintain environmental health and local groundwater quality. The central objective function is mainly composed of three sub-functions:

1. Minimal energy consumption for the water supply system. If v(i) is the water consumption coefficient of a given water source, regarding the consumed power needed to supply each unit volume of water, then differences in v(i) reflect the different energy consumption rates of various water sources. The total power consumption for the entire city's water supply can be expressed by the following equation:

$$\text{Min OBJ1} = \text{Min} \sum_{i=1}^{n} V(i)Q(i) \tag{1}$$

where $n$ is the number of different water sources; $V(i)$ is the water consumption coefficient of each water source in kWh/m³; and $Q(i)$ is the quantity of water supplied by source $i$, in m³.

Concerning constraint conditions, within a given unit of time, the sum of all water supply sources must be equal to the city's total water consumption, and the quantity of water supplied by each source must be less than its maximum water supply.

$$Q_0 = \sum_{i=1}^{n} Q(i), \quad Q_{MIN}(i) \leq Q(i) \leq Q_{MAX}(i), \quad i = 1, 2, \cdots, n \tag{2}$$

Thus, the quantity of water supply from each source at its minimal energy consumption level can be obtained.

2. Minimal water shortage rate. Levels of water availability or differences in access rates and shortages across various water departments can indirectly reflect economic benefits to the recipients.

$$\text{Min OBJ2} = \sum_{j} \alpha_j \cdot \left( MC_{ty}^{j} \cdot \alpha_c + MI_{ty}^{j} \cdot \alpha_i + ME_{ty}^{j} \cdot \alpha_e + MA_{ty}^{j} \cdot \alpha_a + MR_{ty}^{j} \cdot \alpha_r \right) + \lambda \cdot XMIN \qquad \forall ty, j \tag{3}$$

where weight $\alpha_j$ and $\alpha_c$ represent the weight of the urban residential water supply; MC represents shortages of urban residential water; $\alpha_i$ represents the weight of the industrial water supply (including for the tertiary industry and construction); MI represents shortages of urban industrial water (including the tertiary industry and construction); $\alpha_e$ represents the weight of water used for environmental purposes, while ME represents the shortages of water used for environmental purposes; $\alpha_a$ represents the weight of the agricultural water supply (also including water used in the forestry, husbandry, and fishing industries) and MA represents shortages of rural agricultural water; and $\alpha_r$ represents the weight of the rural residential water supply, while MR represents shortages of rural residential water. Furthermore, in order to prevent issues of seasonal shortages of water used for agricultural purposes, the principle of "broad and shallow depletion" shall be followed to minimize disruptions to the water supply across dimensions of both time and space. The weighting factor $\lambda$ represents the uniformity of agricultural depletion, and IN represents the resulting degree of uniformity in the depletion of agricultural resources.

3. Prioritization of water supply. The advantages and disadvantages of various water sources are identified, along with their energy consumption per unit of water supplied, and the supply thresholds of each water source are analyzed.

$$\text{Max OBJ3} = \sum_{j} \alpha_{sur} \cdot \left( SC_{ty}^{j} + SI_{ty}^{j} + SE_{ty}^{j} + SA_{ty}^{j} + SR_{ty}^{j} \right) + \sum_{j} \alpha_{div} \cdot \left( TC_{ty}^{j} + TI_{ty}^{j} + TE_{ty}^{j} + TA_{ty}^{j} + TR_{ty}^{j} \right) + \\ \sum_{j} \alpha_{grd} \cdot \left( GC_{ty}^{j} + GI_{ty}^{j} + GE_{ty}^{j} + GA_{ty}^{j} + GR_{ty}^{j} \right) + \sum_{j} \alpha_{rec} \cdot \left( RI_{ty}^{j} + RE_{ty}^{j} + RA_{ty}^{j} \right) \quad \forall ty, j \tag{4}$$

where $\alpha_{sur}$ represents the weight of the water supply of surface water (including locally produced water), $\alpha_{div}$ represents the weight of externally transferred water, $\alpha_{grd}$ represents the weight of local groundwater supplies, and $\alpha_{rec}$ represents the weight of the reclaimed water supply. For the definitions of other parameters, see Table 2.

**Table 2.** Definitions of relevant variables.

| SC | Surface water for urban residential use | SI | Surface water for industrial use |
|---|---|---|---|
| SE | Surface water for urban environmental use | SA | Surface water for agricultural use |
| SR | Surface water for rural residential use | TC | Transferred water supply for urban residential use |
| TI | Transferred water supply for industrial use | TE | Transferred water supply for urban environmental use |
| TA | Transferred water supply for agricultural use | TR | Transferred water supply for rural residential use |
| GC | Groundwater for urban environmental use | GI | Groundwater for industrial use |
| GE | Groundwater for urban environmental use | GA | Groundwater for agricultural use |
| GR | Groundwater for rural residential use | RI | Reclaimed water for industrial use |
| RE | Reclaimed water for urban environmental use | RA | Reclaimed water for agricultural use |

*3.3. Network Diagram*

This paper uses the city- and county-level administrative districts within the Jinan metropolitan area as "calculation regions", ensuring a geographic basis for calculating the allocation of water resources. These include the five districts of the city (Shizhong, Lixia, Tianqiao, Huaiyin, and Licheng) as well as Changqing District, Pingyin County, Jiyang County, Shanghe County, and Zhangqiu city, amounting to a total of six regions (if Jinan's districts are counted as one). A total of 22 small, medium-sized, and major water source projects within these regions were selected as the basic engineering nodes for the network diagram of the proposed water resource allocation system. According to the data transmission status and hydraulic connection between each water conservancy project and its statistical monitoring unit, all engineering nodes and their statistical monitoring unit were integrated into the general water network. For the resulting network map, see Figure 3.

*3.4. Analysis Process*

The four major water sources in Jinan city were assessed, including transferred water from the Yangtze and Yellow Rivers, local surface water, groundwater, and reclaimed water. Different methods of optimization and configuration patterns were then examined to simultaneously consider the three issues of water supply safety, spring water protection, and environmental mediation. The water resource allocation mechanism was taken as the primary means for pursuing these goals, with a three-configuration multiple-cycle iterative method adopted to obtain the optimal solution. After time-consuming monthly-adjusted calculations and comparative analyses, a multi-dimensional equilibrium configuration scheme for water resource allocation was finally suggested. The steps of the iteration were as follows:

Step 1: The various water resources and sources of Jinan city were assessed, as were the development and utilization of these water resources, the current state of social and economic development and the future development orientations, and other factors. This allows for the analysis of urban water supply risks, water supply/demand plans, and different development modes and water saving scenarios under different planning conditions, as well as the production of forecasts for water supply and demand.

Step 2: By analyzing and generalizing the topological relationships and supply–consumption relationships between different water supply projects of different water supply networks in the research area, a network diagram of water resources available for use by potential allocation systems was produced, water resource allocation models were constructed, and various parameters and variables within the model were precisely defined. A variety of models were established under different objectives and various constraints, and a database of water resource allocation models was established. Based on a given water supply capacity (including supplies of reclaimed water, surface water, externally transferred water, and extractable groundwater W0, or juxtaposed as j = 0) and predicted estimates of water demand, a long series of monthly-adjusted calculations were performed using these water resource allocation models. After defining a current year and setting the balance of the supply

and demand of water resources at different estimated levels, a single optimal configuration result that best fits the objectives and constraints of the model would be produced.

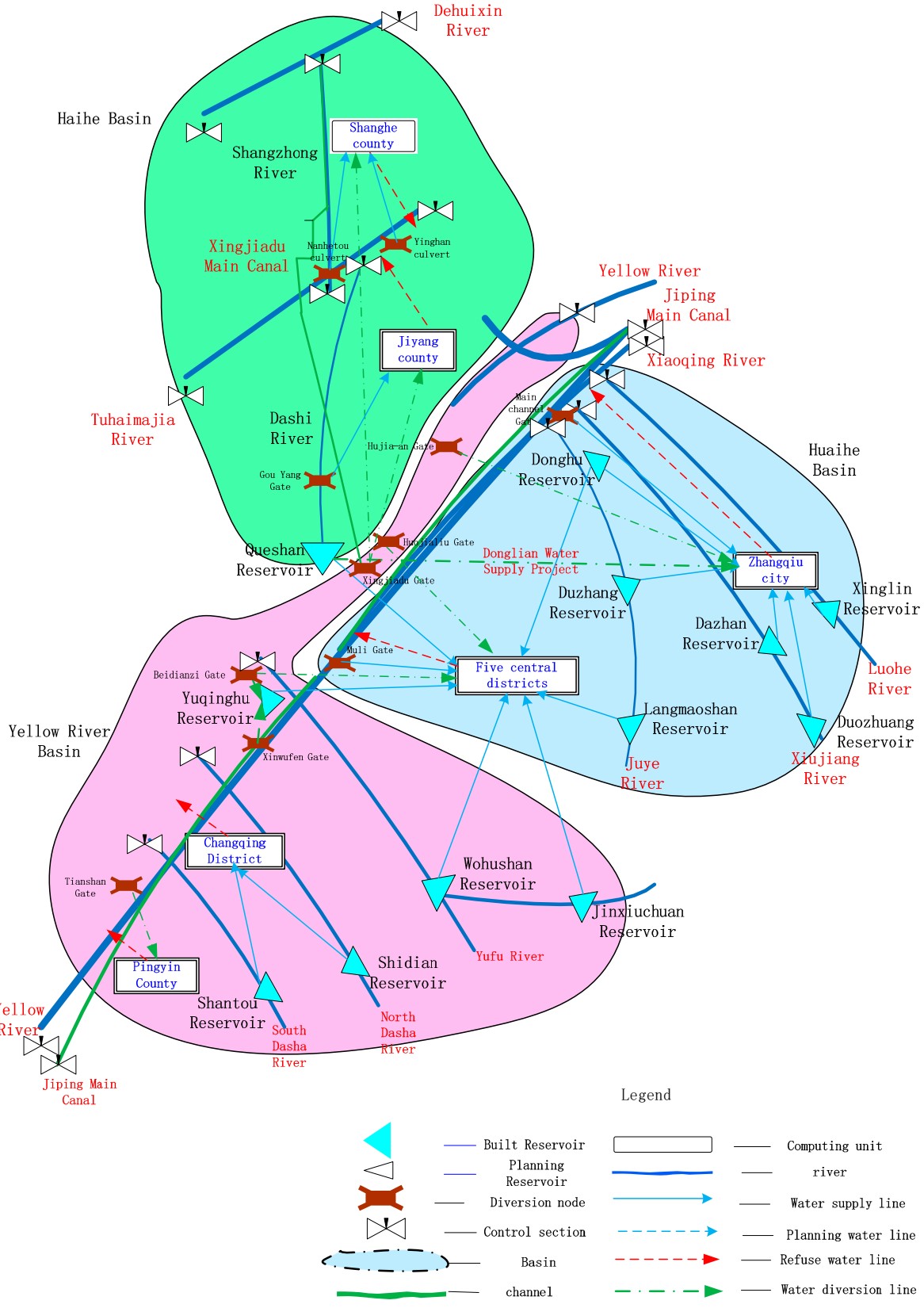

**Figure 3.** Network diagram of the allocation system for Jinan city water Resources.

Step 3: The water supply quantity of each water source and the water consumption of each industry were input into the water equilibrium model, and the total groundwater recharge quantity of each equilibrium unit was obtained, resulting in the determination of the quantity of groundwater recovered $W_j$ (where j is the number of iterations, j = 0, 1, 2, …, n). If $\left|W_{j+1} - W_j\right|/W_j \leq \varepsilon$, the iterative calculation was stopped and its result output; otherwise, the process returned to Step 2, with j set to j + 1, repeating the previous operational processes until their conditions were met and the secondary configuration result was obtained.

Step 4: A numerical model for groundwater was constructed according to the hydrogeological conceptual model. This numerical model was then used to measure a series of data to identify and verify the model. The supply quantity of each water source and the water consumption of each industry were then applied to the numerical groundwater model. This model was then used to hypothesize future periods of surplus, equilibrium, and shortages to predict future trends in groundwater dynamics and to control the groundwater level within position hj (with j being the number of iterations used on the model: j = 1, 2, …, n).

Step 5: If Hlower limit ≤ hj ≤ Hupper limit, then the process proceeded to Step 6; otherwise, the quantity of extracted groundwater Qj was rationally adjusted until Hlower limit ≤ hj ≤ Hupper limit was satisfied. At this time, Wj = Qj, and should return to the beginning of Step 5, with j changed to j + 1, and the previous operational process was repeated.

Step 6: If $\left|W_{j+1} - W_j\right|/W_j \leq \varepsilon$, then the process proceeded to Step 7. If not, it returned to Step 4, with j changing to j + 1. The previous operational process was then repeated until the above conditions were met and the three configuration results were received.

Step 7: The iterative calculation was ceased and the result of the general urban water resource allocation model was output, including the calculation results of the groundwater, other water sources, and water consumption rates of various industries in the five districts of Jinan city as well as Changqing District, Zhangqiu city, Pingyin County, Jiyang County, and Shanghe County.

Step 8: Add the energy consumption constraint in the water resource allocation model. The minimization of energy consumption during use is considered, while balancing the allocation of water supply and demand in different areas of Jinan as well as accounting for future variations. The iterative calculation ceased, and the final result of the water resource allocation model based on minimum energy consumption was output.

## 4. Results and Discussion

### 4.1. Unit Energy Consumption (UEC) of Water Extraction

This study quantifies and compares the energy consumption benchmarks for the supply, consumption, drainage, and treatment of different water sources in Jinan [24–27] and comprehensively organizes the energy consumption levels of each form of water resource utilization.

#### 4.1.1. Power Consumption during the Intake of Surface Water

The Jinan Administration has not yet measured the city's energy consumption of its surface water intake processes. The Yuqing Reservoir, which is the main reservoir supplying water to Jinan, lies about 8 km from the nearest water plant, and the distance from the water plant to the Wohushan Reservoir is about 39 km. Water is taken from each reservoir according to current needs. The average distance at which surface water is extracted is set at 15 km, and the efficiency of the given pump is 75%. The energy consumption of Jinan's surface water intake process is thus 0.069 kWh/m$^3$.

#### 4.1.2. Power Consumption during External Water Transfers

The efficiency of the external intake pump is 75%. Jinan draws its externally sourced water primarily from the Yellow River and the Yangtze River; apart from that, water is supplied to the city via three reservoirs (1.2 million m$^3$/day): Queshan, Yuqing Lake, and

Donghu. This water is used mainly for industrial purposes. In 2020, 568 million m$^3$ will ultimately be coming from the Yellow River, while 100 million m3 will be sourced from the Yangtze River; this will increase to 200 million m$^3$ by 2030. The remote water transfer facility consumes 0.0045 kWh/m$^3$ for every 1 km that this water is transported. The average distance from the Yellow River to the reservoirs is 68 km, and that from the Yangtze River to the reservoirs is about 600 km. Thus, the energy consumption for Jinan's intake of water from the Yellow River is 0.306 kWh/m$^3$, while the energy consumption of extracting and transporting Yangtze River water to Jinan as surface water is 2.7 kWh/m$^3$.

### 4.1.3. Power Consumption during the Extraction of Groundwater

The energy consumption of the groundwater extraction process is related mainly to the well depth and pumping station efficiency. According to our survey, the average depth of groundwater pumped from Shandong Province (on the northern plain of the Yellow River) is 68 m, with the deepest well being 224 m and the shallowest being only 6 m. Considering that pump efficiency tends to range from 60–85%, this study takes 65% as a conservative average. According to the average values of the variables in Shandong, the power consumption of groundwater extraction in Jinan is 0.285 kWh/m$^3$.

### 4.1.4. Power Consumption of Water Reclamation

For the purposes of this study, an on-site inspection was conducted at Jinan Guangda Sewage Company's Jinan Branch Sewage Treatment Plant. This site was found to have a power consumption of 0.25 to 0.30 kWh/m$^3$ during treatment, consumed mainly by fans and pumps, and both the quality and quantity of water to be treated had impacts on energy consumption. The power consumption of the urban sewage collection and treatment process was 0.29 kWh/m$^3$. Membrane separation technologies such as ultrafiltration are used to treat the effluents at the secondary sewage treatment plant and produce reclaimed water, and the energy consumption of this process is about 0.1 kWh/m$^3$. The total power consumption rate of producing reclaimed water is therefore 0.39 kWh/m$^3$.

### 4.1.5. Power Consumption of Desalination

There is currently no seawater desalination project in Jinan. According to the results of this research, the energy consumption of desalinating brackish water is 0.3–1.4 kWh/m$^3$, and the power consumption of the desalinization and water production processes measured for one instance is 1.4 kWh/m$^3$.

### 4.1.6. Power Consumption of Urban Water Supply, Treatment, and Distribution

The intake and transport of water to a waterwork account for 80–90% of the power consumption at a given plant; the treatment process itself consumes little energy. The Yuqing Water Plant is the largest water supply plant in Jinan city. The Yuqing Water Plant provides about 80% of Jinan's tap water. After conducting an on-site investigation at the Yuqing Water Plant, local power consumption per unit of water supplied in 2016 was found to be between 0.167 and 0.170 kWh/m$^3$; most of this power consumption is attributed to electricity production, with about 90% being used by booster pumps, while backwashing and sewage discharge accounted for most of the remaining 10%. The power consumption intensity of treating and distributing water in Jinan is 0.17 kWh/m$^3$.

### 4.2. Model Results and Discussion

After the analysis of the water resources supply and demand balance, it was shown that the total water demand in Jinan city in 2030 will be 2.339 billion m$^3$. Based on the current state of development and utilization of water resources, as well as an analysis of the potential for local water resource development and utilization, a comprehensive comparison of technological and economic factors was performed, allowing for the development of multiple sets of development and utilization plans and the forecasting of available water supplies. The results of this analysis are shown in Table 3.

**Table 3.** Analysis and calculation results of available water in Jinan. Units: million m$^3$.

| Administrative Division | Year(s) | Surface Water | | | Groundwater | | | External Water Transfer | Alternative Water Sources | Total Water Supply Capacity | | |
|---|---|---|---|---|---|---|---|---|---|---|---|---|
| | | 50% | 75% | 95% | 50% | 75% | 95% | | | 50% | 75% | 95% |
| Five central districts | 2019 | 194.7 | 168.0 | 109.4 | 166.2 | 157.9 | 149.6 | 317.0 | 75.6 | 753.5 | 718.5 | 651.6 |
| | 2030 | 194.7 | 168.0 | 109.4 | 166.2 | 157.9 | 149.6 | 364.0 | 231.7 | 956.6 | 921.5 | 854.6 |
| Changqing District | 2019 | 67.1 | 47.1 | 29.7 | 75.1 | 71.3 | 67.6 | 3.0 | 0.5 | 145.7 | 121.9 | 100.8 |
| | 2030 | 67.1 | 47.1 | 29.7 | 75.1 | 71.3 | 67.6 | 13.0 | 27.4 | 182.6 | 158.8 | 137.7 |
| Zhangqiu City | 2019 | 79.1 | 76.4 | 28.3 | 207.8 | 197.4 | 187.0 | 82.0 | 1.0 | 369.8 | 356.8 | 298.3 |
| | 2030 | 79.1 | 76.4 | 28.3 | 207.8 | 197.4 | 187.0 | 105.0 | 49.0 | 440.8 | 427.7 | 369.3 |
| Pingyin County | 2019 | 37.5 | 33.2 | 20.1 | 79.7 | 75.7 | 71.7 | 25.0 | 0.1 | 142.3 | 134.0 | 116.9 |
| | 2030 | 37.5 | 33.2 | 20.1 | 79.7 | 75.7 | 71.7 | 25.0 | 12.2 | 154.3 | 146.1 | 129.0 |
| Jiyang County | 2019 | 22.0 | 8.6 | 6.2 | 95.4 | 90.6 | 85.8 | 169.0 | 2.1 | 288.5 | 270.4 | 263.2 |
| | 2030 | 22.0 | 8.6 | 6.2 | 95.4 | 90.6 | 85.8 | 189.0 | 18.1 | 324.4 | 306.3 | 299.1 |
| Shanghe County | 2019 | 27.2 | 7.6 | 7.0 | 107.4 | 102.0 | 96.6 | 72.0 | 0.6 | 207.1 | 182.2 | 176.2 |
| | 2030 | 27.2 | 7.6 | 7.0 | 107.4 | 102.0 | 96.6 | 72.0 | 17.2 | 223.8 | 198.8 | 192.9 |
| Total | 2019 | 427.5 | 340.9 | 200.7 | 731.5 | 694.9 | 658.3 | 668.0 | 80.0 | 1906.9 | 1783.7 | 1606.9 |
| | 2030 | 427.5 | 340.9 | 200.7 | 731.5 | 694.9 | 658.3 | 768.0 | 355.5 | 2282.5 | 2159.3 | 1982.5 |

The results of each iterative cycle of water supply calculations for all water sources are shown in Table 4. Energy demand associated with water is growing faster than water demand itself (due to more energy-intensive water sources). Therefore, Jinan needs strategies to prioritize less energy-intensive water sources. Based on the available natural resources and their development and utilization status in the research area and taking into account the future national economic demand for water as well as different water resource allocation patterns and resource management objectives at different development stages, this study determined the total amount of groundwater to be withdrawn to meet national water demands in accordance with the corresponding groundwater level. The proposed water resource allocation model is based on minimizing energy consumption during use, while balancing the allocation of water supply and demand in different areas of Jinan as well as accounting for future variations. In 2020, the total water supply in Jinan will be 1.917 billion m$^3$, including the surface water supply of 396 million m$^3$, the groundwater supply of 731 million m$^3$, the reclaimed water supply of 206 million m$^3$, and the external water supply of 584 million m$^3$. The total water shortage will be 36 million m$^3$, with a water shortage rate of 1.8%. In 2030, the total water supply in Jinan is estimated to be 2.214 billion m$^3$, including the surface water supply of 427 million m$^3$, the groundwater supply of 731 million m$^3$, the reclaimed water supply of 355 million m$^3$, and the external water supply of 70 million m$^3$. The total water shortage will be 24 million m$^3$, with a water shortage rate of 1.1%. The city will thus have basically balanced its supply and demand for water.

**Table 4.** Results of multiple iterations of water supply calculations for each water source in 2030. Units: million m$^3$.

| Model | Surface Water | Ground Water | Reclaimed Water | Yellow River Water | Yangtze River Water | Total |
|---|---|---|---|---|---|---|
| GM * | 426.9 | 663.5 | 355.5 | 468.0 | 200.0 | 2214.0 |
| LM * | 426.9 | 731.5 | 355.5 | 568.0 | 132.1 | 2214.0 |

Note: * The general urban water resource allocation model is referred to as GM; the water resource allocation model is referred to as LM.

The results show that in 2030, the total water supply in Jinan is estimated to be 33.7% higher than the 2014 levels, but energy consumption will also have increased by 58.5%. Considering the optimal allocation of energy consumption, the utilization of high-energy-draw sources of water (such as externally sourced water) should be reduced, and the amount of groundwater extraction should be rationally increased while maintaining a balance between the supply and demand of both local springs and groundwater. The energy consumption of all aspects of water resource allocation increased by 44.2%, as shown in Table 5.

**Table 5.** Energy consumption status quo in the study area [1]. Units: million m$^3$, %, million kWh.

| Administrative Division | 2019 | | | 2030 | | | |
|---|---|---|---|---|---|---|---|
| | WS | WSR | EC0 | WS | WSR | EC1 | EC2 |
| Five central districts | 688.0 | 0.5 | 281.4 | 956.6 | 0.7 | 447.3 | 429.7 |
| Changqing District | 132.0 | 0.4 | 28.4 | 175.8 | 0.0 | 60.3 | 44.9 |
| Zhangqiu city | 330.0 | 3.8 | 99.7 | 370.4 | 1.9 | 130.3 | 130.3 |
| Pingyin County | 114.0 | 2.8 | 35.7 | 130.9 | 0.8 | 50.7 | 41.5 |
| Jiyang County | 306.0 | 1.3 | 143.0 | 345.3 | 0.9 | 207.0 | 176.1 |
| Shanghe County | 195.0 | 2.1 | 49.6 | 235.0 | 2.5 | 115.1 | 97.3 |
| Total | 1764.0 | 1.7 | 637.6 | 2214.0 | 1.1 | 1010.6 | 919.8 |

Notes: [1] WS: water supply; WSR: water shortage rate; EC0: energy consumption (2019); EC1: energy consumption under triple equilibrium; EC2: energy consumption under usage constraints.

## 5. Conclusions

In this paper, the issue of energy consumption in the allocation of urban water resources was discussed. Focused on Jinan city and its major water resources—local surface water, groundwater, transferred water, and reclaimed water—the energy consumption per unit water supply of the four water sources was estimated, and the future water demand of different units in the region was also analyzed. Based on existing and planned water resource projects, the area's future water availability was estimated. We used the allocation model with the smallest water scarcity coefficient and the water resource allocation based on the lowest energy consumption to analyze the water demand and calculated water consumption, respectively. The results show that:

1. Jinan's local water use is relatively energy-efficient and that there is an inherent conflict between the health status of local springs and the utilization of groundwater.
2. If energy consumption is not considered, it is feasible to use reclaimed water and externally sourced water to resolve water shortage issues. However, the growth rate of energy consumption in Jinan's case is much higher than the increase in water supply.
3. Power consumption during the extraction of groundwater is less than external water transfers.
4. Considering the optimal allocation of energy consumption, the utilization of water transfers should be reduced, and the amount of groundwater extraction should be rationally increased.

According to the scheme based on the water resource allocation model with the lowest energy consumption, it will be more energy efficient, and transferred water and reclaimed water can be used to balance water supply and demand. In the future, the water resource allocation model based on minimizing energy consumption will be used. However, it is necessary to consider the energy involved in reclaiming and transporting water from distant external sources when making decisions from this aspect. In the future, we will further analyze the relationship between water supply and energy consumption and continue to develop energy- and water-saving means of developing water resources and utilization methods that draw upon multiple water sources.

**Author Contributions:** Conceptualization, Z.Y.; methodology, Z.Y.; formal analysis, Z.Y.; writing—original draft preparation, Z.Y.; writing—review and editing, Z.Y. and G.M.K.; visualization, J.D. and L.C.; supervision, G.M.K. All authors have read and agreed to the published version of the manuscript.

**Funding:** This research was funded by the National Natural Science Foundation of China (Grant No. 51509266).

**Data Availability Statement:** The data presented in this study are available on request from the corresponding author.

**Conflicts of Interest:** The authors declare no conflict of interest.

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
