# Peer review of "City Water Resource Allocation Considering Energy Consumption in Jinan, China"

_water, doi:10.3390/w15163016_

Round 1
Reviewer 1 Report
Title: Unless the authors explain how their method is more generalizable, the study is limited to Jinan and I suggest they include “Jinan, China” in the title to give it the identity of a case study.
General: I like the idea of accounting for energy intensity step by step through an entire urban water supply process. This paper does just that. It has some flaws that could be addressed with a major revision.
Abstract: One important finding needs to appear here: local groundwater is less energy intensive than imported surface water, which suggests groundwater should be preferred (at least for energy reasons).
Lines 31–42: Another big U.S. study is by Chini and Stillwell (https://doi.org/10.1002/2017WR022265).
Line 69: I suggest a new paragraph starting here that better states the research problem. What are the authors going to do?
Table 1: More justification is needed here. Where did these judgments come from? For example, why is the Yangtze River source to Jinan not stressed by climate change, while Yellow River and other surface waters are?
Lines 95–103: These sources must appear fully in the References.
Lines 106–108: Please cite these sources.
Line 123: Something is missing here.
Figure 3: The figure is illegible. Please increase resolution.
Section 4.1 (lines 293–320): This appears to be literature review and belongs earlier, not under Results and Discussion.
Section 4.1.1: The authors have not provided enough information to substantiate “thus 0.069 kWh/m3.” Total dynamic head is needed. Also, is 75% efficiency assumed or known?
Section 4.1.2: State the basis for 0.0045 kWh/m3, e.g., “The facility consumed ___ kWh for pumping __ m3 over ___ years.” Same for all following sections.
General: A helpful reference value for pumping is 0.00272 kWh/m3 per m of head, as derived by Sowby and Krieger (https://doi.org/10.22541/au.168727973.37485149/v1), through which I verified several of the authors’ calculations.
General: I would like to see more discussion of how the energy intensity values for Jinan compare to other places and processes. There is some literature (at least from the U.S.) available for this purpose. See Sowby and Burian (already citied) Chini and Stillwell (above), Hanna et al. (https://doi.org/10.2175/106143017X15131012153176), and various reports from EPRI (2002, 2007, 2009, and 2013).
Line 421: I believe here, and throughout, the authors mean “water demand” when they say “water supply.” If so, please adjust.
Lines 421–423: This is an important point: Energy demand associated with water is growing faster than water demand itself (due to more-energy-intensive water sources). Therefore, Jinan needs strategies to prioritize less-energy-intensive water sources.
General: Can the authors offer recommendations for tracking energy performance of the Jinan water system, filling measurement gaps (e.g., surface water intake, lines 322–323), and sharing data? See Sowby et al. (https://doi.org/10.1002/awwa.1233)
References: Much needs to be done to clean up this list. Author names are missing; punctuation, capitalization, and abbreviation are inconsistent; and CrossRef links all point to the same article. This is sloppy.
Author Response
According to the reviewer’s comments, we have revised the manuscript extensively. If there are any other modifications we could make, we would like very much to modify them and we really appreciate your help.

Reviewer 2 Report
After carefully reading the entire paper, I suggested releasing it under the title "City Water resource allocation considering energy consumption" after major revision.
1. More clarity and rewriting are needed for the introduction, materials, and technique.
2. It is necessary to investigate the effects of the nearby water resources and the electricity consumption.
The manuscript need to grammar check
Author Response

(The authors gave the same response as above.)

Round 2
Reviewer 1 Report
I am satisfied with the revisions.